# How fast are viruses spreading in the wild?

**Simon Dellicour** [1,2,3]*, **Paul Bastide**[4], **Pauline Rocu**[5], **Denis Fargette**[6], **Olivier J. Hardy**[3,7], **Marc A. Suchard**[8,9,10], **Stéphane Guindon**[5], **Philippe Lemey**[2]

**1** Spatial Epidemiology Lab (SpELL), Université Libre de Bruxelles, Brussels, Belgium, **2** Department of Microbiology, Immunology and Transplantation, Rega Institute, KU Leuven, Leuven, Belgium, **3** Interuniversity Institute of Bioinformatics in Brussels, Université Libre de Bruxelles, Vrije Universiteit Brussel, Brussels, Belgium, **4** IMAG, Université de Montpellier, CNRS, Montpellier, France, **5** Department of Computer Science, Laboratoire d'Informatique, de Robotique et de Microélectronique de Montpellier, CNRS and Université de Montpellier, Montpellier, France, **6** PHIM Plant Health Institute, Université de Montpellier, IRD, CIRAD, INRAE, Institut Agro, Montpellier, France, **7** Laboratoire d'Evolution Biologique et Ecologie, Faculté des Sciences, Université Libre de Bruxelles, Brussels, Belgium, **8** Department of Human Genetics, David Geffen School of Medicine, University of California, Los Angeles, California, United States of America, **9** Department of Biostatistics, Fielding School of Public Health, University of California Los Angeles, Los Angeles, California, United States of America, **10** Department of Computational Medicine, David Geffen School of Medicine, University of California Los Angeles, California, United States of America

* simon.dellicour@ulb.be

**Data Availability Statement:** R scripts related to the analyses based on simulated and real datasets are all available, along with the associated input/output files, at https://github.com/sdellicour/dispersal_capacities. Continuous phylogeographic

## Abstract

Genomic data collected from viral outbreaks can be exploited to reconstruct the dispersal history of viral lineages in a two-dimensional space using continuous phylogeographic inference. These spatially explicit reconstructions can subsequently be used to estimate dispersal metrics that can be informative of the dispersal dynamics and the capacity to spread among hosts. Heterogeneous sampling efforts of genomic sequences can however impact the accuracy of phylogeographic dispersal metrics. While the impact of spatial sampling bias on the outcomes of continuous phylogeographic inference has previously been explored, the impact of sampling intensity (i.e., sampling size) when aiming to characterise dispersal patterns through continuous phylogeographic reconstructions has not yet been thoroughly evaluated. In our study, we use simulations to evaluate the robustness of 3 dispersal metrics — a lineage dispersal velocity, a diffusion coefficient, and an isolation-by-distance (IBD) signal metric — to the sampling intensity. Our results reveal that both the diffusion coefficient and IBD signal metrics appear to be the most robust to the number of samples considered for the phylogeographic reconstruction. We then use these 2 dispersal metrics to compare the dispersal pattern and capacity of various viruses spreading in animal populations. Our comparative analysis reveals a broad range of IBD patterns and diffusion coefficients mostly reflecting the dispersal capacity of the main infected host species but also, in some cases, the likely signature of rapid and/or long-distance dispersal events driven by human-mediated movements through animal trade. Overall, our study provides key recommendations for the use of lineage dispersal metrics to consider in future studies and illustrates their application to compare the spread of viruses in various settings.

simulations and dispersal statistics were respectively conducted and computed using the R package "seraphim" available at https://github.com/sdellicour/seraphim (see also the updated "seraphim" tutorial on the estimation of dispersal statistics available at: https://github.com/sdellicour/seraphim/blob/master/tutorials/).

**Funding:** SD and PL acknowledge support from the European Union Horizon 2020 project MOOD (grant agreement n°874850). SD also acknowledges support from the *Fonds National de la Recherche Scientifique* (F.R.S.-FNRS, Belgium; grant n°F.4515.22), from the Research Foundation — Flanders (*Fonds voor Wetenschappelijk Onderzoek — Vlaanderen*, FWO, Belgium; grant n°G098321N), and from the European Union Horizon 2020 project LEAPS (grant agreement n° 101094685). PR's internship at the University of Montpellier was founded by the I-SITE MUSE through the Key Initiative "Data and Life Sciences". MAS and PL acknowledge support from the European Union's Horizon 2020 research and innovation programme (grant agreement no. 725422-ReservoirDOCS), from the Wellcome Trust through project 206298/Z/17/Z, and from the National Institutes of Health grants R01 AI153044, R01 AI162611 and U19 AI135995. PL also acknowledges support from the Research Foundation — Flanders (*Fonds voor Wetenschappelijk Onderzoek — Vlaanderen*, FWO, Belgium; grants n°G0D5117N and G051322N). The funders had no role in study design, data collection and analysis, decision to publish, or preparation of the manuscript.

**Competing interests:** The authors have declared that no competing interests exist.

**Abbreviations:** AIV, avian influenza virus; BRW, Brownian random walk; HPD, highest posterior density; IBD, isolation-by-distance; MCC, maximum clade credibility; MDC, mean diffusion coefficient; MLDV, mean lineage dispersal velocity; RRW, relaxed random walk; WDC, weighted diffusion coefficient; WLDV, weighted lineage dispersal velocity.

## Introduction

Unravelling the spatiotemporal dynamics of pathogenic spread constitutes a long-standing challenge in epidemiology. When designing and implementing intervention strategies to mitigate outbreaks or endemic circulation of pathogens, evaluating the speed at which they spread and circulate within host populations can be crucial. While geo-localised infectious cases can be used to model and quantify the wavefront progression of an outbreak during its expansion phase (see e.g., [1,2]), their analysis usually provides little information about the routes taken by the underlying transmission chains. Genomic sequencing of fast-evolving pathogens, however, offers the possibility to infer evolutionary relationships among sampled cases. Estimated through a phylogenetic tree, such inferred evolutionary links can for instance provide key insights into the dispersal history and dynamics of a transmission chain [3]. This can be achieved through phylogeographic inference, either performed in discrete [4–6] or continuous space [7,8], which basically consists in mapping time-scaled phylogenetic trees of fast-evolving organisms such as RNA viruses (see e.g., [9,10]) or, sometimes, DNA viruses (see e.g., [11,12]) or bacteria (see e.g., [13,14]).

In particular, continuous phylogeographic inference allows estimation of a spatially explicit reconstruction of the dispersal history of the lineages leading to the sampled genomic sequences. The reconstructions in two-dimensional space can subsequently be exploited to gain valuable information about the dispersal dynamics of circulating lineages [15], including their capacity to disperse in space and time. However, for a given outbreak, phylogeographic reconstructions are generally conducted on genomic sequences from a limited proportion of cases, with sampling intensities that can vary considerably among studies.

In the context of these heterogeneous sampling intensities, the goal of the present study is to assess the usefulness of dispersal metrics to evaluate and compare the dispersal dynamic of viral lineages. Specifically, we first aim to assess the robustness of 3 metrics — a lineage dispersal velocity, a diffusion coefficient, and an isolation-by-distance (IBD) signal metric — to the sampling intensity, i.e., the number of genomic samples considered in the phylogeographic analysis. In addition, we aim to exploit the metrics identified as robust to the sampling intensity (i.e., sampling size) to compare the dispersal capacity across a variety of viruses of public and one health importance (e.g., Lassa, rabies, avian influenza, or West Nile viruses).

At relatively large scales, dispersal dynamics of viruses primarily spreading within human populations are generally better captured by discrete diffusion processes involving the rapid exchange of viral lineages among remote locations — a typical pattern for remote locations mainly connected by the air traffic network (see e.g., [16,17]). This is for instance the case for seasonal human influenza virus H3N2, for which dispersal at the global scale has been demonstrated to be better predicted by international air travel than geographic distances [18]. While continuous phylogeographic analyses can also be performed to reconstruct the dispersal history of viruses spreading in human populations (see e.g., [19–21]), our study focuses on viruses circulating in animal populations to target dispersal processes that are relatively strongly dependent on geographic distance. Such dispersal patterns can in essence be modelled through a continuous diffusion process, such as the relaxed random walk (RRW) diffusion model implemented in the software package BEAST 1.10 [22], which is often used for continuous phylogeographic inference.

## Results and discussion

We here assess 3 distinct dispersal metrics: the weighted lineage dispersal velocity (WLDV) [23], the weighted diffusion coefficient (WDC) [24], and an IBD signal metric that we here propose to estimate as the Pearson correlation coefficient between the patristic and log-

transformed great-circle geographic distances computed for each pair of tip nodes (which is similar to the metric estimated by Pekar and colleagues [25]). The great-circle distance is the geographic distance between points on the surface of the Earth and the patristic distance is the sum of the branch lengths that link 2 tip nodes in a phylogenetic tree. While the WLDV aims at measuring the distance covered per unit of time across phylogenetic history, the WDC measures the "diffusivity" [24], i.e., the velocity at which inferred lineages invade a two-dimensional space:

$$WLDV = (\Sigma_i d_i)/(\Sigma_i t_i) \text{ and } WDC = (\Sigma_i d_i^2)/(\Sigma_i 4t_i),$$

where $d_i$ and $t_i$ are, respectively, the great-circle distance and the time elapsed on the $i$-th branch of the phylogeny. The concept of IBD was first introduced by Sewall Wright [26] to describe the accumulation of local genetic differences under geographically restricted dispersal [27,28]. IBD analyses generally consist in determining the strength and significance of the relationship between a genetic and geographic distance between individuals [29]. The IBD signal metric considered here aims to measure to what extent phylogenetic branches are spatially structured or the tendency of phylogenetically closely related tip nodes to be sampled from geographically close locations. Note that this third metric is therefore not based on a continuous phylogeographic reconstruction but directly on a time-scaled phylogeny and sampling locations.

To evaluate the robustness of these 3 dispersal metrics to the sampling intensity (i.e., sampling size), we have conducted phylogeographic simulations either based on a Brownian random walk (BRW) or a RRW diffusion process. For this purpose, we have first implemented 2 distinct forward-in-time simulators, both jointly simulating time-scaled phylogenies and the dispersal history of their branches on an underlying geo-referenced grid. At each time step of the BRW simulation, both the longitudinal and latitudinal displacements of evolving lineages are randomly drawn from a Gaussian distribution, while in the RRW simulations such longitudinal and latitudinal displacements are randomly drawn from a Cauchy distribution before randomly rotating the resulting displacement around its origin (see the Materials and methods section for further detail). An example of a geo-referenced phylogeny simulated through a Brownian diffusion process is displayed in Fig 1 (see Figure A in S1 Text for the example of a RRW simulation). To investigate the potential impact of the phylogenetic tree inference on the different dispersal statistic estimates, we have also conducted alternative phylogeographic simulations based on tree topologies simulated under a coalescent model considering a population with a constant size (Figure B in S1 Text). The aim of all these simulations is to obtain continuous phylogeographic reconstructions that can subsequently be subsampled to artificially generate data sets that would have been obtained through various levels of sampling intensity.

For each diffusion model considered, we have simulated 50 continuous phylogeographic processes of more than 500 tips and then subsampled the resulting locations to only keep 500, 450, 400, 350, 300, 250, 200, 150, 100, and 50 tip locations (see the Materials and methods section for further detail). We subsequently computed the 3 dispersal metrics on all the resulting data sets to explore the impact of the sampling size on their estimates (Fig 2). Although associated with slightly more variability when estimated on the smaller data sets, both the diffusion coefficient and IBD signal metrics appear to be robust to the sampling intensity (Fig 2). This is however not the case for the lineage dispersal velocity metric for which estimates drastically decrease with the number of tips (Fig 2). While Fig 2 reports the results obtained with the continuous phylogeographic simulations based on a Brownian diffusion process, we reach the same conclusions based on the results obtained by the simulations following an RRW diffusion process (Figure C in S1 Text) or when tree topologies are simulated under a coalescent instead of a birth-death model (Figure D in S1 Text).

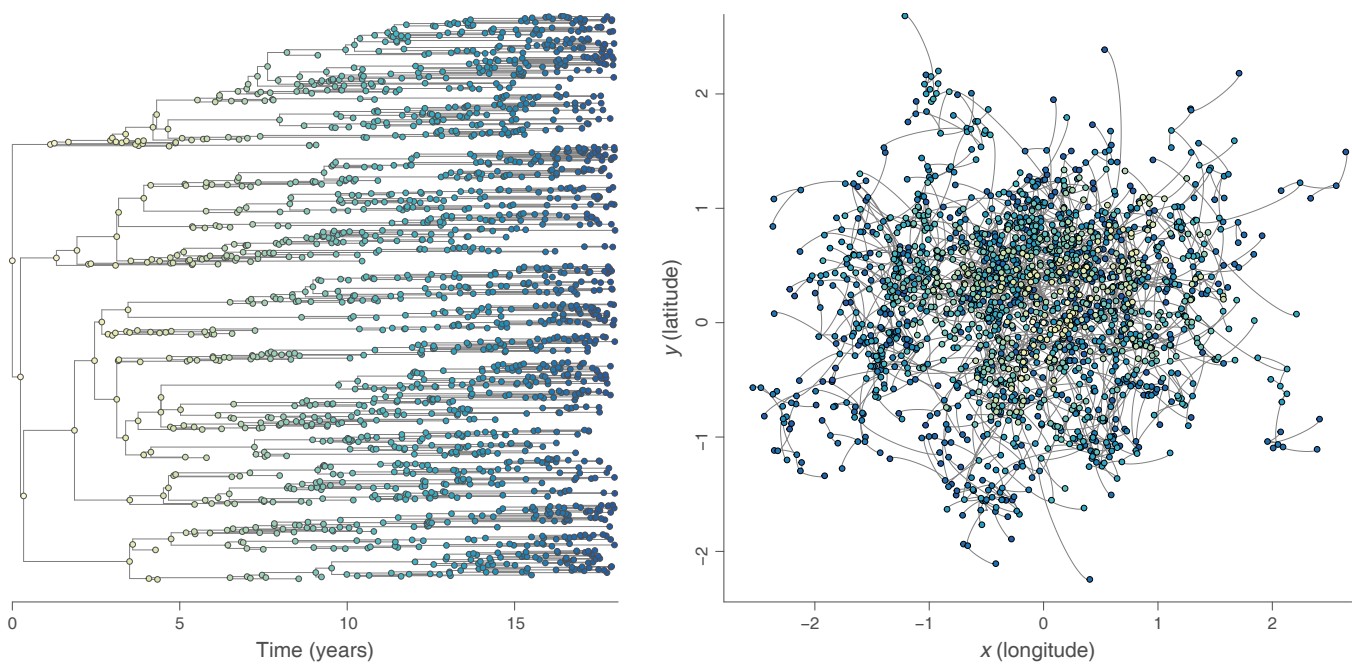

**Fig 1. Example of a continuous phylogeographic simulation based on a BRW diffusion process.** Both graphs display the phylogenetic tree sampled during a unique simulation, with its time-scaled visualisation in the left panel and its mapped visualisation in the right panel. Tree nodes are coloured according to time, with internal and tip nodes coloured according to their time of occurrence and collection time, respectively. The data underlying this figure can be found in https://doi.org/10.5281/zenodo.13984927. BRW, Brownian random walk.

In addition to the 3 main metrics investigated above, we have also used our simulations to assess the robustness of alternative metrics, such as different ways to define the correlation coefficient measuring the IBD signal (Figure E in S1 Text) or unweighted versions of the lineage dispersal velocity and diffusion coefficient (Figure F in S1 Text). Overall, we reach the same conclusion: while the lineage dispersal velocity estimates are impacted by the sampling

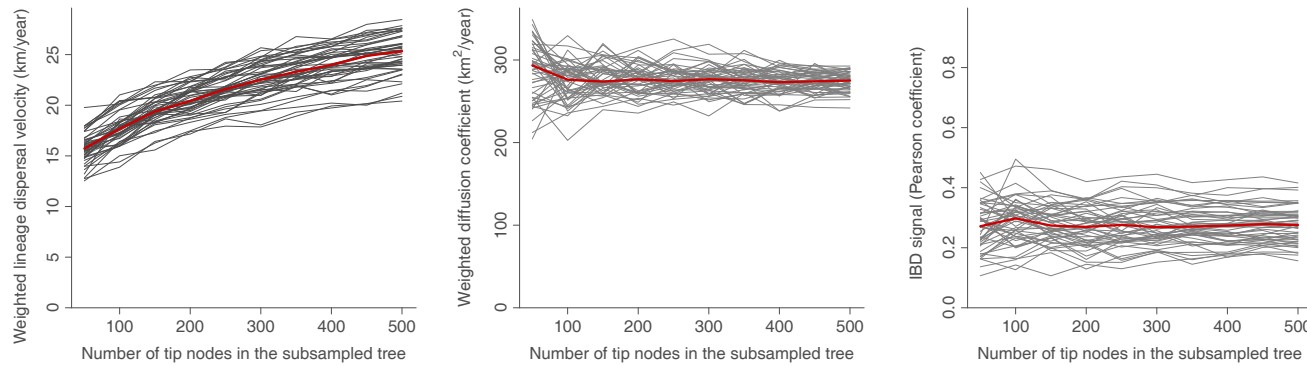

**Fig 2. Robustness of lineage dispersal metrics to the sampling intensity.** We here report 3 dispersal statistics estimated on 50 geo-referenced phylogenetic trees simulated under a Brownian diffusion process: the WLDV (km/year), the WDC (km²/year), and the IBD signal has been estimated by the Pearson correlation coefficient ($r_P$) between the patristic and log-transformed great-circle geographic distances computed for each pair of tip nodes. Each specific tree is represented by a specific grey curve obtained when re-estimating the dispersal metric on subsampled versions of the tree, i.e., subsampled trees obtained when only randomly keeping 500, 450, 400, 350, 300, 250, 200, 150, 100, and 50 tip nodes; and the red curve indicate the median value across all simulated trees. The data underlying this figure can be found in https://doi.org/10.5281/zenodo.13984927. IBD, isolation-by-distance; WDC, weighted diffusion coefficient; WLDV, weighted lineage dispersal velocity.

intensity, this does not seem to be the case for the diffusion coefficient estimates and all investigated IBD signal metrics.

From a mechanistic perspective, the lack of robustness of the lineage dispersal velocity metric is expected from a Brownian motion process, which has paths with infinite variation on any interval (see e.g., Corollary 2.17 in [30]). This implies that, the more densely sampled the trajectory is, the larger the WLDV becomes, making the estimation of an average speed sampling inconsistent. In contrast, the Brownian motion has finite quadratic variation (see e.g., Proposition 2.16 in [30]), so that the WDC, which uses quadratic distances instead of linear ones, should not be sensitive to the number of samples. To illustrate these theoretical expectations, we consider the case of a strict one-dimensional Brownian diffusion process of position $X$ along a time interval equal to 1, with $N$ being the total number of regular time intervals $dt = 1/N$, $d_i$ the distance travelled during time interval $i$, $X_i$ the position at the end of interval $i$, and $X_{pa(i)}$ the position at the end of the previous time interval. Under a strict Brownian process, we have that $X_i - X_{pa(i)}$ is a Gaussian random variable, so that the distance covered on one interval $d_i = \sqrt{\left(X_i - X_{pa(i)}\right)^2} = |X_i - X_{pa(i)}|$ is distributed as a half normal variable:

$$d_i = |X_i - X_{pa(i)}| \sim N_+(0, \sigma^2 dt) \text{ and } E(d_i) = \sigma\sqrt{dt}\sqrt{2/\pi},$$

which allows to determine the expected value for the WLDV as follows:

$$E(WLDV) = \frac{\sum_{i=1}^{N} E(d_i)}{\sum_{i=1}^{N} dt} = \frac{\sum_{i=1}^{N}(\sigma\sqrt{dt}\sqrt{2/\pi})}{\sum_{i=1}^{N} dt} = \sigma\sqrt{\frac{2}{\pi}}\frac{\sum_{i=1}^{N}\sqrt{dt}}{\sum_{i=1}^{N} dt} = \sigma\sqrt{\frac{2}{\pi}}\frac{\sum_{i=1}^{N}\sqrt{1/N}}{1}$$

$$= \sigma\sqrt{\frac{2}{\pi}}\frac{N}{\sqrt{N}} = \sigma\sqrt{\frac{2}{\pi}}\sqrt{N}.$$

This indicates that lineage dispersal velocity depends, on average, on the number of time intervals $N$ and thus the duration of these time intervals: the shorter these intervals, the larger the estimate. Because the phylogenetic branch lengths (durations) will on average decrease with the number of tip nodes, this implies, by extension, that the lineage dispersal velocity depends on the number of tip nodes included in the tree, as also illustrated by our simulations (Fig 2). Further, the quadratic distance $d_i^2$ is in this special case a scaled $\chi^2$ random variable with one degree of freedom, so that:

$$\frac{d_i^2}{\sigma^2 dt} = \frac{\left(X_i - X_{pa(i)}\right)^2}{\sigma^2 dt} \sim \chi^2(k=1), E(d_i^2) = \sigma^2 dt \text{ and}$$

$$E(WDC) = \frac{\sum_{i=1}^{N} E(d_i^2)}{\sum_{i=1}^{N} 4dt} = \frac{\sum_{i=1}^{N} \sigma^2 dt}{4\sum_{i=1}^{N} dt} = \sigma^2\frac{\sum_{i=1}^{N} dt}{4} = \frac{\sigma^2}{4}.$$

Therefore, WDC is proportional to the squared net lineage displacement per unit of time ($\sigma^2$) and does not depend on $N$, so that it can be used to compare the dispersal capacities of different pathogens. Note that this relationship between the WDC and $\sigma^2$ is only valid in this toy model of a one-dimensional Brownian diffusion where the process is measured at all time steps $dt$. In a phylogenetic context, the geographic position of internal nodes of the tree is unknown, and the WDC is typically computed with inferred ancestral positions, so that the relationship between the WDC and $\sigma^2$ is not that simple in general and depends on the ancestral reconstruction method being used. The WDC metric should hence not be used as a replacement for classical estimators of the diffusion coefficient of a Brownian model, either based on the (restricted) maximum likelihood [31] or, in the context of this study, Bayesian inference [7].

In addition to the simulations conducted with a birth-death or coalescent approach, we have also conducted continuous phylogeographic simulations by simulating RRW diffusion processes along the branches of an empirical phylogenetic tree, i.e. the maximum clade credibility (MCC) tree previously obtained from the phylogeographic analysis of the West Nile virus in North America [32] (Figure G in S1 Text; see also the Materials and methods section for further detail). The specific aim of these additional simulations is to assess the robustness of the dispersal metrics to the sampling intensity, as well as to a scenario of sampling bias, but this time also considering the uncertainty associated with phylogenetic inference. As illustrated in Figure H in S1 Text, our results indicate that when explicitly considering the uncertainty associated with the inference of the time-scaled tree topologies, the impact of various scenarios of sampling intensity on the dispersal metrics is less evident. Indeed, the posterior distribution of the dispersal metric estimates appears to be often quite large, sometimes preventing a clear distinction of any impact of sampling intensity on these estimates. Furthermore, in order to take into account the uncertainty related to phylogenetic inference, this additional simulation framework required performing a continuous phylogeographic inference based on each subset of geo-referenced tips sampled from the original simulation for generating various scenarios of sampling intensity (see the Materials and methods section for further detail). Because of this additional inference step, it seems that the random selection of sequences has often more impact on the dispersal metric estimates than the number of samples that were subsampled, especially when the sampling size decreases (Figure H in S1 Text). The later result thus highlights the stochastic impacts of the sampling itself, independently of the impact of the sampling size. Finally, we also note that use of an RRW instead of a BRW diffusion model to conduct these continuous phylogeographic simulations could also contribute to the relatively large variability observed among simulations; a phenomenon that was already observed when comparing the results obtained when considering either a BRW or an RRW model to conduct the simulations under a birth-death process (Fig 1 and Figure C in S1 Text).

As introduced above, we have also used these simulations of RRW diffusion processes along empirical tree branches to estimate the dispersal metrics from simulations based on a scenario of sampling bias where only a restricted area of the spread has been sampled (see the Materials and methods section for further detail). Such a sampling bias scenario has previously been shown to strongly affect continuous phylogeographic reconstructions [33]. Again, our results based on this simulation scenario and explicitly considering phylogenetic uncertainty do not indicate a clear impact of sampling bias on the estimates (Figure H in S1 Text). So with this simulation framework, we do not manage to distinguish a clear difference between the dispersal metrics estimated for the scenarios of various sampling intensity and sampling bias. The impact of such spatial sampling bias on the outcomes of continuous phylogeographic inference has been more thoroughly investigated in previous studies [8,33]. For example, Kalkaukas and colleagues have explored the impact of 3 sampling bias scenarios on the estimation of the squared net lineage displacement per unit of time ($\sigma^2$, referred to as the "diffusion parameter" in their work [33]), which is directly related to a metric like WDC investigated here (see above). Their analyses reveal that all considered scenarios of sampling bias — a central sampling bias, a diagonal sampling bias, and a one-side sampling bias scenario similar to the one explored in this study — lead to an underestimation of the diffusion parameter. Further confirmed by additional investigations of sampling bias scenarios by Guindon and De Maio [8], these results thus indicate that sampling bias can lead to an underestimation of the dispersal capacity as quantified by a metric like WDC.

In the second part of our study, we use the 2 dispersal metrics identified above as the most robust to the sampling intensity — the WDC and IBD signal — to quantify and compare the dispersal capacity of viruses spreading in animal populations. We here target viruses of public

and one health importance and for which comprehensive genomic data sets have at some point been exploited to conduct continuous phylogeographic reconstructions that we have managed to retrieve. The selected data sets encompass viruses associated with a broad range of host species (Table A in S1 Text): moles (Nova virus), rodents (Puumala, Lassa, and Tula viruses), deer (Powassan virus), poultry and waterfowl (avian influenza virus (AIV)), domestic pigs (Getah virus and porcine deltacoronavirus), cattle (lumpy skin disease virus), (migratory) bird species (West Nile virus), as well as raccoons, skunks, bats, and dogs (rabies virus). By definition, the transmission cycles of the 3 considered arboviruses involve transmission events through biting by infected arthropods such as ticks of the *Ixodes* and *Dermacentor* genera for the Powassan virus, as well as mosquito species of the *Culex* genus for the West Nile virus and also *Aedes* genus in the case of the Getah virus.

Our results highlight substantial differences among the data sets, both in terms of diffusion coefficient and IBD signal (Fig 3). By far, the 2 virus data sets associated with the lowest diffusion coefficient are the Nova virus and Puumala virus genomic sequences collected in Belgian mole (*Talpa europaea*) and bank vole (*Myodes glareolus*) populations, respectively [49,50]. The very limited diffusion coefficient estimated for these 2 hantaviruses is coherent with the limited dispersal capacity of their host species. The Lassa virus data set [48] is next in the ranking with a WDC estimated at approximately 40 $km^2$/year (Fig 3 and Table A in S1 Text). Lassa virus is an arenavirus that also circulates in a rodent species: the Natal multimammate mouse (*Mastomys natalensis*). Interestingly, these 3 first data sets all reveal a relatively high IBD signal reflecting an important phylogeographic structure (Fig 3 and Table A in S1 Text).

A large fraction of genomic data sets result in diffusion coefficient estimates between 100 and 2,000 $km^2$/year (Table A in S1 Text). They range from the Powassan virus data set [47] from the United States (WDC = ~130 $km^2$/year) to the avian influenza virus (AIV) H3N1 data set [37] corresponding to a specific Belgian outbreak (~1,800 $km^2$/year). In between these 2 diffusion coefficient estimates, we find the diffusion coefficients estimated from a series rabies virus data sets (Fig 3 and Table A in S1 Text): ~250 $km^2$/year for the bat rabies virus data set from eastern Brazil [46], almost ~600 $km^2$/year for the raccoon [44] and skunk [43] rabies virus data sets from the USA, ~700 $km^2$/year for the bat rabies virus data set in Argentina [42], ~1,100 $km^2$/year for the dog rabies data sets, respectively, from the Yunnan province (China) and northern Africa [41], ~1,400 $km^2$/year for the bat rabies dataset from Peru [39], and ~1,600 $km^2$/year for the dog/wild carnivore rabies data set from Iran [38]. Interestingly, although the diffusion coefficient estimated for the bat rabies data set from Peru is comparatively high, rabies viruses circulating in bats do not seem to be associated with a notably higher diffusion coefficient compared to other non-flying host species. Furthermore, it has previously been hypothesised that the relatively higher diffusion coefficient estimated for the dog rabies virus data sets compared to other rabies virus datasets could be the result of occasional rapid (long-distance) dispersal events driven by human-mediated movements [41,51]. Of note, we observe an important heterogeneity among the IBD signals estimated for the different rabies virus data sets, the IBD signal metric being centred on zero for the dog rabies data set from Yunnan or close to zero for the raccoon one from the USA, and higher than ~0.65 for the dog rabies data set from northern Africa and the bat rabies data set from Peru.

We subsequently identify 3 data sets with diffusion coefficient estimates ranging from ~20,000 to ~26,000 $km^2$/year including an AIV H5N1 data set from the Mekong region [36] and 2 genomic data sets of viruses infecting domestic pig populations in China [34,35]: the Getah virus, a re-emerging arbovirus China, and the porcine deltacoronavirus. Similarly to the hypothesis formulated for the dog rabies data sets such as the one from northern Africa, the relatively high diffusion coefficient for those 2 pig virus data sets may potentially be explained by long-distance dispersal through animal trade. It is also interesting to note the very limited

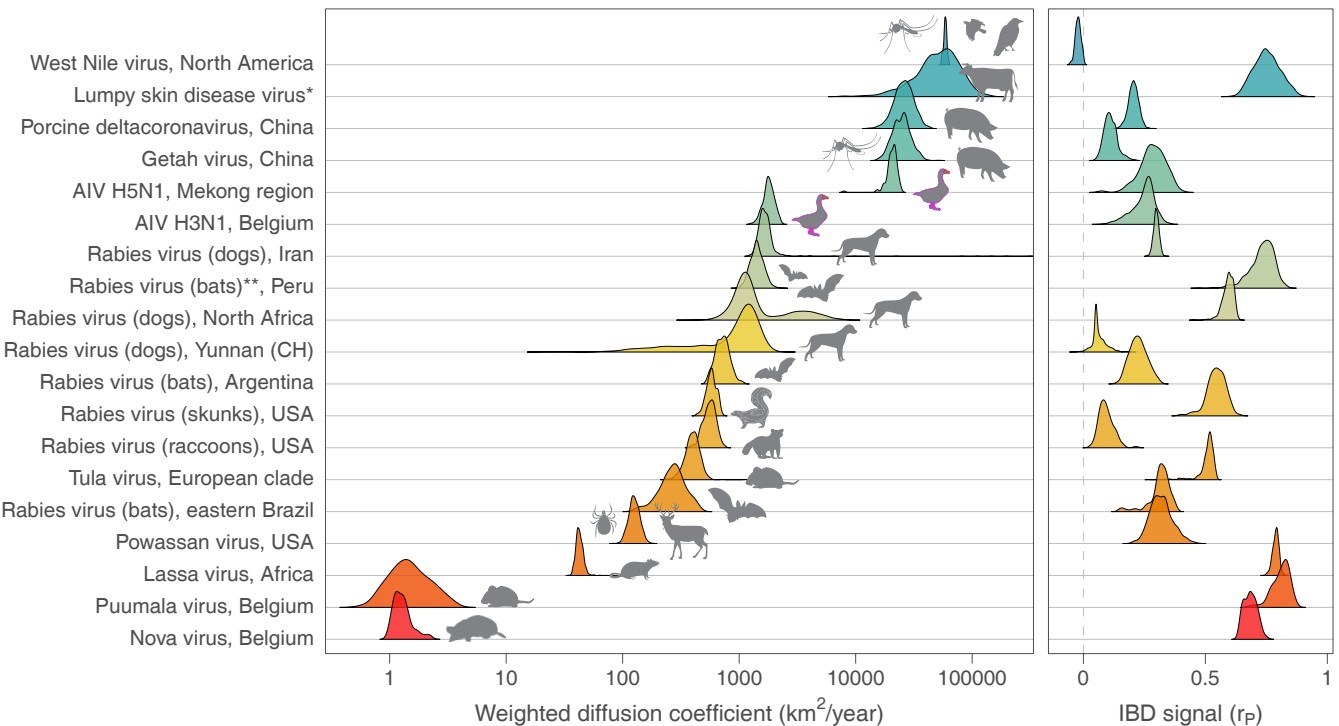

**Fig 3. Comparison of dispersal metrics estimated for different genomic data sets of viruses spreading in animal populations.** Specifically, we here report posterior estimates obtained for 2 metrics estimated from trees sampled from the posterior distribution of a Bayesian continuous phylogeographic inference: the WDC and the IBD signal estimated by the Pearson correlation coefficient ($r_P$) between the patristic and log-transformed great-circle geographic distances computed for each pair of virus samples. We report the posterior distribution of both metrics estimated through continuous phylogeographic inference for the following data sets: West Nile virus in North America [32], Lumpy skin disease virus [11], Porcine deltacoronavirus in China [34], Getah virus in China [35], AIV in the Mekong region [36] and H3N1 in Belgium [37], rabies virus (dogs) in Iran [38], rabies virus (bats) in Peru [39], rabies virus (dogs) in northern Africa [40,41], rabies virus (bats) in Argentina [42], rabies virus (skunks) in the USA [43], rabies virus (raccoons) in the USA [44], Tula virus in central Europe [45], rabies virus (bats) in eastern Brazil [46], Powassan virus in the USA [47], Lassa virus in Africa [48], Puumala virus in Belgium [49], and Nova virus in Belgium [50]. (*) Estimates based on the analysis of the wild-type strains (see [11] for further detail); (**) estimates based on the combined analysis of lineages L1 and L3. See also Table A in S1 Text for the related 95% HPD intervals and number of samples associated with each data set. The data underlying this figure can be found in https://doi.org/10.5281/zenodo.13984927. AIV, avian influenza virus; HPD, highest posterior density; IBD, isolation-by-distance; WDC, weighted diffusion coefficient.

or absent IBD signal estimated for the deltacoronavirus and Getah virus data sets, respectively, which is consistent with the notable spatial inter-mixing of lineage dispersal events that would be induced by animal trade across the Chinese territory [34,35].

Finally, with posterior median estimates >50,000 km$^2$/year, the North American West Nile virus [32] and lumpy skin disease virus [11] data sets yield the highest diffusion coefficient estimates. Upon first detection in 1999 in New York City, the West Nile virus spread through the North American continent, carried by numerous (migratory) birds and reaching the West Coast of the United States within a period of only 3 years [52]. Lumpy skin disease virus was first detected in 1929 in northern Rhodesia (now Zambia) and initially circulated within the African continent. During the last decades, it has however spread eastward and to southeastern Europe [53]. Transmitted by a variety of arthropod vectors (ticks, mosquitoes, and biting flies), the main hosts of this virus are cattle and buffalos while the role of other wild animals in its transmission and spread remains uncertain [53]. The continuous phylogeographic reconstruction of the lumpy skin disease virus dissemination considered here is a global one involving long-distance lineage dispersal events among continents. Also in this case, the high diffusion coefficient estimated for this data set is potentially related to the transport of infected

cattle to naive regions [11]. It is also worth noting that the IBD signal estimated for the West Nile virus and lumpy skin disease virus data sets are drastically different, being centred on zero for the former and approximately 0.8 for the latter. This result really reflects the phylogeographic patterns reconstructed for both outbreaks, the West Nile virus one displaying an important amount of longitudinal, and to some extent latitudinal, back and forth lineage dispersal events during the endemic phase of the North American outbreak, and the lumpy skin disease virus spread mainly consisting in long-distance dispersal events establishing more local transmission chains.

Furthermore, the relatively large diffusion coefficient estimated for the West Nile virus data set appears to be partially due to the expansion phase of this outbreak: when computing a distinct diffusion coefficient for the expansion and endemic phases — here approximated by considering phylogenetic branches occurring before or after 2004 [32], respectively — we obtain an estimate that is between 2 and 3 times higher for the expansion phase (Table A in S1 Text), illustrating the impact of the distinction of those epidemic phases on the metric. In the case of the West Nile virus spread across the North American continent, the invasion was in part driven by long-distance dispersal events, which can transform the spread of a pathogen from a wave-like to a fast "metastatic" growth pattern [54]. Overall, the comparison of the estimates based on the West Nile virus data set highlights that a metric like the diffusion coefficient can notably be impacted by the epidemic phase of the considered outbreak, an expansion phase being more likely to be associated with higher estimates than an endemic phase.

Finally, while we have previously established that the 2 dispersal metrics estimated on those empirical data sets appear to be robust to the sampling intensity, they remain impacted by sampling bias like any outcome of a phylogeographic inference. As reported in Figure I in S1 Text, we see that the sampling maps of the different empirical data sets display an important variability in terms of sampling patterns, with a varying degree of sampling site clustering. The associated sampling bias is however difficult to evaluate in practice given that the actual spatial distribution of individuals/infectious cases is frequently unknown, especially when studying pathogens spreading in wild animal populations. We can nevertheless clearly speculate that infected areas were likely under or even unsampled for a good number of those data sets. As established in previous works [8,33], we cannot exclude that such sampling bias could lead to a significant underestimation of the dispersal capacity as measured by the WDC metric.

## Limitations, conclusions, and perspectives

Our study comes with a number of limitations. First, while our simulation framework explores the impact of sampling intensity (i.e., heterogeneity in the absolute sampling effort) and, to a more limited extent, sampling bias (i.e., spatial heterogeneity in the sampling effort), it does not explicitly explore the impact of border effects, which can also affect the target statistics. Furthermore, Layan and colleagues have illustrated that the extent of the study area can impact the diffusion coefficient estimates [55]. As an interpretation, they put forward that "larger sampling areas are expected to be associated with higher probabilities to sample long-distance dispersal events that will, on average, more likely correspond to fast dispersal events than short-distance dispersal events" [55], which is also in line with the underestimation of the diffusion parameter $\sigma^2$ highlighted by Kalkaukas and colleagues when considering a "one-side" sampling bias scenario [33], i.e., a scenario where only a restricted part of the distribution area is actually sampled. As detailed in Table A in S1 Text, we however see that the different datasets analysed here encompass a large range of study area sizes that are not necessarily following the trend of diffusion coefficient estimates.

Second, our simulation framework does not involve the generation of actual genomic sequences. From a computational perspective, adding a genomic sequence simulation step would also have drastically increased the computation time and resources required to conduct each Bayesian phylogeographic analysis. This would indeed imply running no less than 1,000 continuous phylogeographic analyses (corresponding to both Brownian and RRW simulations, times 50 simulations, times the generation of 10 subsampled data sets with a decreasing number of tip nodes). To circumvent this limitation, we have considered an additional simulation framework to still consider the phylogenetic uncertainty but while avoiding the simulation and time-consuming analyses of genomic sequences. As detailed in the Materials and methods section, this simulation framework is made of 3 distinct steps: (i) a continuous phylogeographic simulation along an empirical time-scaled phylogeny; (ii) the subsampling (according to a scenario of sampling intensity/bias) of the tip nodes whose position has been simulated on the map and, for each resulting data set; (iii) the continuous phylogeographic inference based on the simulated coordinates of the selected tip nodes while integrating over posterior trees retrieved from the phylogeographic inference previously conducted on the actual data set. Interestingly, due to a relatively higher impact of phylogenetic uncertainty or the subsampling itself of tip nodes, these additional simulations do not allow to distinguish any clear trend on the impact of sampling intensity/bias on the estimates of the different dispersal metrics evaluated here.

Spatially explicit phylogeographic reconstructions are relatively widely applied thanks to popular implementations of the RRW model [7,15,56], but there is a need for robust metrics allowing a direct comparison of lineage dispersal dynamics for different viral variants/outbreaks or even distinct viruses. Our simulations highlight that, among the 3 different kinds of dispersal metrics investigated here, only 2 of them appear to be robust to the sampling intensity: the diffusion coefficient and the IBD signal metrics, the first one being estimated from continuous phylogeographic inference and the second one being directly estimated from a (time-scaled) phylogeny and sampling locations. The lineage dispersal velocity estimated from continuous phylogeographic inference on the other hand increases with the number of samples in the phylogeographic analysis. We subsequently confirmed that this trend can be explained by a mechanistic dependence of this metric on the number of genomic sequences (tip nodes) considered in the analysis.

The 2 metrics identified as robust are measuring complementary aspects of the overall dispersal pattern: while the diffusion coefficient allows estimating dispersal capacity, the IBD signal metric will quantify to what extend the geo-referenced phylogenies are spatially structured, i.e., the tendency for phylogenetically related lineages to evolve in a close vicinity. In terms of readability, the diffusion coefficient (WDC) can also be used to estimate the quadratic mean — also called root mean square — of the effective displacement by unit of time, which is equal to $s = \sqrt{4WDC}$. Furthermore, if the mean time needed for a pathogen to infect successive hosts by contagion (a kind of generation time, $T$) was known, the root mean square dispersal distance between infection locations of 2 successive hosts could be estimated as $s_T = \sqrt{4WDC*T}$.

In the second part of our study, we have taken advantage of these 2 robust metrics to compare the dispersal pattern of viral lineages informed by the available continuous phylogeographic reconstruction of a large range of published data sets of genomic sequences. Our results confirm and highlight an important heterogeneity in the dispersal capacity of viruses primarily spreading in animal populations, directly reflecting the dispersal capacity and/or the human-mediated movement of the main host species carrying those viruses. Future studies applying continuous phylogeographic analyses to new genomic data sets can further build on the comparison of dispersal metric estimates that we initiated here. In practice, the GitHub

repository referenced below can be used to add additional data sets in the comparison reported in Fig 3 and Table A in S1 Text, with the perspective to further unveil and understand the velocity at which viruses can circulate in various host populations.

In our view, the field of phylogeography, or phylodynamic in general, applied to fast-evolving pathogens such as RNA (and DNA) viruses faces a series of (new) challenges. A first challenge lies in the recent and impressive increase in the amount of genomic information that can be exploited in phylodynamic analyses. Driven by technological developments in next-generation sequencing [57–59] and important national budgets unlocked for genomic surveillance in the context of the COVID-19 pandemic, the number of geo-referenced genomic sequences available for conducting phylogeographic reconstruction of some viruses of public health importance has exploded, which is particularly true for SARS-CoV-2 with >16.5 million full genomes available on the GISAID platform to date (March 2024, https://gisaid.org/). Unfortunately, Bayesian approaches that jointly infer phylogeny and phylogeographic history to accommodate uncertainty in both processes reach their computational limit when analysing datasets consisting of a few thousand genomes. While recent developments have been dedicated to the phylogenetic inference of large-scale genomic data sets (see e.g., [60,61]), facing computational limitations will continue to be a timely challenge in the development of phylodynamic approaches aiming to look below the tip of the iceberg by exploiting the large amount of pathogen genetic information at our disposal in order to get quantitative insights into or test hypotheses on ongoing epidemics. Another challenge likely lies in the perspective to go beyond the retrospective aspect of phylodynamic analyses and feed the knowledge obtained through such reconstructions into predictive frameworks.

## Materials and methods

We conducted continuous phylogeographic simulations based on both a Brownian (BRW) and a relaxed (RRW) random walk diffusion process on an underlying geo-referenced grid (raster) with a resolution of 0.5 arcmin, and also along phylogenetic trees inferred from actual genomic data in the case of the RRW diffusion process.

### Continuous phylogeographic simulations based on a birth-death process

We first implemented a forward-in-time birth-death approach to simulate the dispersal history of viral lineages over 20 years while considering a time step of one day. Specifically, each simulation started from an index case placed in the centre of the grid. At each time step, each ongoing lineage had (i) a first probability to give birth to another lineage ($p_{birth}$ = 0.6 per year, so 0.0016 per day); and (ii) a second probability to be "sampled" ($p_{sampling}$ = 0.4 per year, so 0.0011 per day), i.e., to stop evolving and leave the simulation. Of note, we defined a sampling window so that evolving lineages could not be sampled during the first 2 years of the simulation/outbreak. In between the probability to give birth and the probability to be sampled, ongoing lineages also had the opportunity, at each time step, to move on the grid. This displacement was defined by a two-steps procedure. In the BRW simulations, the longitudinal and latitudinal displacements were first sampled by respectively drawing an horizontal $d_x$ and vertical $d_y$ displacement value from a Gaussian distribution with mean equal to zero and standard deviation here arbitrarily set to 0.0125, and in the RRW simulations, the longitudinal and latitudinal displacements were first sampled by respectively drawing an horizontal $d_x$ and vertical $d_y$ displacement value from a Cauchy distribution with a position parameter set to zero and a scale parameter set to $2.5 \times 10^{-4}$. In both cases, the resulting displacement on the map was then randomly rotated around the previous location point. In particular for the RRW simulations, this stochastic rotation step prevents rare long-distance displacement to only occur

along a latitudinal or longitudinal direction. Overall, we conducted 200 BRW and 200 RRW simulations and retained in each case the first 50 simulations leading to simulated phylogenies with at least 500 tip nodes. Selected phylogeographic simulations were then eventually subsampled to obtain geo-referenced time-scaled phylogenies only including 500, 450, 400, 350, 300, 250, 200, 150, 100, and 50 tip nodes. The script used to conduct the BRW and RRW simulations was implemented in a new function named "simulatorRRW2" and included in the R package "seraphim" [23,62] (the "simulatorRRW1" function corresponding to another simulator previously implemented to conduct RRW simulations along fixed tree topologies [19]).

### Continuous phylogeographic simulations based on a coalescent process

Secondly, in addition to the simulations with phylogenetic trees simulated under a birth-death model, we conducted BRW simulations along trees primarily simulated under a coalescent model. To perform these coalescent simulations, we used the function "rcoal" of the R package "ape." Longitudinal and latitudinal displacements were then simulated along the resulting tree topologies with the "fastBM" function of the R package "phytools" [63]. As for the simulations based on a birth-death model, we conducted 200 simulations and retained in each case the first 50 simulations leading to simulated phylogenies with at least 500 tip nodes, and those phylogeographic simulations were eventually subsampled to obtain geo-referenced time-scaled phylogenies only including 500, 450, 400, 350, 300, 250, 200, 150, 100, and 50 tip nodes.

### Continuous phylogeographic simulations conducted along a tree topology

Thirdly, we also conducted RRW simulations along the branches of an empirical phylogenetic tree: the MCC tree obtained from the continuous phylogeographic reconstruction based on 801 West Nile virus genomes collected between 1999 and 2016 in North America [32] (Figure G in S1 Text). To this end, we used the "simulatorRRW1" function implemented in the R package "seraphim" to conduct RRW simulations along time-scaled tree topologies [19]. Specifically, we simulated distinct forward-in-time RRW diffusion processes along the branches of the considered MCC tree using the precision matrix parameters and root node location inferred by the original continuous phylogeographic analysis, this location having been inferred in the vicinity of New York City [19]. These simulations were either spatially unconstrained or constrained to the contiguous area of the United States, and 5 distinct simulations were conducted for each of these 2 categories of simulations (see Figure G in S1 Text for the example of one spatially constrained simulation). Again, we subsequently subsampled each simulation to obtain time-scaled phylogenies only including 500, 450, 400, 350, 300, 250, 200, 150, 100, and 50 tip nodes associated with sampling coordinates simulated by the RRW diffusion process along the original MCC tree. In order to investigate the robustness of the dispersal metrics to the sampling intensity while considering the uncertainty associated with the phylogenetic inference, we then performed a new continuous phylogeographic reconstruction based on each resulting data set. To incorporate the uncertainty associated with phylogenetic inference, we performed all phylogeographic reconstructions while integrating over 100 posterior trees retrieved from the phylogeographic inference previously conducted on the actual West Nile virus data set [32]. In this way, we account for tree topology uncertainty when assessing the robustness of the dispersal metrics to the sampling intensity without the need to confront inference from sequence data.

Finally, we also exploited the RRW simulations conducted along a tree topology to explore the impact of 2 scenarios of sampling bias on the robustness of the 3 dispersal metrics considered in our study. For this purpose, we used the simulation procedure described above, but instead of randomly subsampling each simulation to obtain data sets associated with various

sampling intensity, we performed a subsampling corresponding to a scenario of spatial sampling bias where only a restricted area of the spatial distribution of the virus spread was sampled. In the case of the spatially unconstrained simulations, the subsampling was performed to only retain all tip nodes simulated east of the tree root location. In the case of the spatially constrained simulations, the subsampling was performed to only retain all tip nodes simulated further east of the 86th West meridian, which roughly corresponds to a subsampling restricted to the first eastern third of the contiguous USA area that was invaded by the West Nile virus.

## Dispersal statistics estimation

Dispersal statistics were all computed using the "spreadStatistics" function in the R package "seraphim" [23,62], which now also allows the estimation of the IBD signal computed as the Pearson correlation ($r_P$) between the patristic and log-transformed great-circle geographic distances computed for each pair of virus samples. 95% highest posterior density (HPD) intervals were computed with the "hdi" function of the R package "HDInterval", and the ridgeline plots were generated using the "ridgeline" R package. As mentioned above, in complement to the WLDV and WDC metrics, we also computed and tested unweighted versions — the mean lineage dispersal velocity (MLDV) [23] and mean diffusion coefficient (MDC) [15]:

$$MLDV = \frac{1}{n}\sum_{i=1}^{n}\left(\frac{d_i}{t_i}\right) \text{ and } MDC = \frac{1}{n}\sum_{i=1}^{n}\left(\frac{d_i^2}{4t_i}\right),$$

where $n$ is the number of phylogenetic branches, $d_i$ the geographic (great-circle) distance, and $t_i$ the time elapsed on the $i$-th branch of the phylogeny.

## Supporting information

**S1 Text.**
(PDF)

## Acknowledgments

We are grateful to two anonymous reviewers for their constructive comments, as well as to Richard Neher for his useful feedback on a previous version of this study.

## Author Contributions

**Conceptualization:** Simon Dellicour, Paul Bastide, Olivier J. Hardy, Stéphane Guindon, Philippe Lemey.

**Data curation:** Simon Dellicour.

**Formal analysis:** Simon Dellicour, Paul Bastide, Pauline Rocu, Olivier J. Hardy.

**Funding acquisition:** Simon Dellicour.

**Investigation:** Simon Dellicour, Paul Bastide, Pauline Rocu, Denis Fargette, Olivier J. Hardy, Stéphane Guindon.

**Methodology:** Simon Dellicour, Paul Bastide, Olivier J. Hardy, Stéphane Guindon, Philippe Lemey.

**Project administration:** Simon Dellicour.

**Resources:** Pauline Rocu, Denis Fargette, Olivier J. Hardy, Marc A. Suchard, Stéphane Guindon, Philippe Lemey.

**Software:** Simon Dellicour, Marc A. Suchard, Philippe Lemey.

**Supervision:** Simon Dellicour, Philippe Lemey.

**Validation:** Simon Dellicour, Paul Bastide, Olivier J. Hardy, Marc A. Suchard, Stéphane Guindon, Philippe Lemey.

**Visualization:** Simon Dellicour.

**Writing – original draft:** Simon Dellicour.

**Writing – review & editing:** Simon Dellicour, Paul Bastide, Pauline Rocu, Denis Fargette, Olivier J. Hardy, Marc A. Suchard, Stéphane Guindon, Philippe Lemey.

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
