## [Editor Report · Decision Letter 0]

7 May 2024

Dear Dr Dellicour, 

Thank you for submitting your manuscript entitled "How fast are viruses spreading in the wild?" for consideration as a Research Article by PLOS Biology.

Your manuscript has now been evaluated by the PLOS Biology editorial staff, as well as by an academic editor with relevant expertise, and I am writing to let you know that we would like to send your submission out for external peer review.

Once your full submission is complete, your paper will undergo a series of checks in preparation for peer review. After your manuscript has passed the checks it will be sent out for review. To provide the metadata for your submission, please Login to Editorial Manager (https://www.editorialmanager.com/pbiology) within two working days, i.e. by May 09 2024 11:59PM.

Kind regards,

Melissa

Melissa Vazquez Hernandez, Ph.D.

Associate Editor

PLOS Biology

---

## [Decision Letter · Decision Letter 1]

8 Jul 2024

Dear Dr Dellicour,

Thank you for your patience while your manuscript "How fast are viruses spreading in the wild?" was peer-reviewed at PLOS Biology. It has now been evaluated by the PLOS Biology editors, an Academic Editor with relevant expertise, and by two independent reviewers, Peter Ralph and Emily Martin. 

In light of the reviews, which you will find at the end of this email, we invite you to revise your work to thoroughly address the reviewers' comments. Both reviewers find the study intriguing but have several concerns that must be addressed before publication. Reviewer #1 requires that you consider spatial sampling and/or location, as well as assess phylogenetic error. Reviewer #2 believes that linking the findings to humans would strengthen the paper. While Reviewer #2's suggestion can be addressed in the discussion, the concerns raised by Reviewer #1 must be implemented in the model and assessed accordingly.

Given the extent of revision needed, we cannot make a decision about publication until we have seen the revised manuscript and your response to the reviewers' comments. Your revised manuscript is likely to be sent for further evaluation by all or a subset of the reviewers.

**IMPORTANT - SUBMITTING YOUR REVISION**

*Re-submission Checklist*

*Published Peer Review*

*PLOS Data Policy*

*Blot and Gel Data Policy*

Sincerely,

Melissa

Melissa Vazquez Hernandez, Ph.D.

Associate Editor

PLOS Biology

REVIEWERS' COMMENTS:

Reviewer #1: Review of "How fast are viruses spreading in the wild?", by Dellicour et al.

This is a straightforward paper that looks at several metrics of dispersal

computed from phylogeographic trees, and applies the metrics to 19 viral

datasets from the literature. So, the first part of the paper evaluates these

metrics on simple branching random walk simulations, and shows that one of them

- the "weighted lineage dispersal velocity" (WLDV) - is sensitive to the sample

size, and hence not a consistent estimator of anything. (The underlying reason

for this is explained with some math.) The second part of the paper evaluates

the remaining two metrics on the viral datasets, and finds a surprising and

interesting array of results.

I am of two minds about this paper. I really like the premise of the paper:

evaluation of these metrics of dispersal seems important, and application to a

wide variety of viral datasets is very interesting, and the diversity of

results found on those datasets is intriguing. In particular, WLDV is used at

least occasionally in the literature, so it seems important to point out its

shortcomings. However, a major shortcoming of the paper is that it does not

explore the effects of sampling *location* (only sample size), which leads me

to question both main conclusions (i.e., that the other two metrics are robust,

and that the comparison between datasets is meaningful). To be clear, I think

this is a useful and interesting paper, but given this substantial gap between

the results and the real world on both fronts, doesn't feel like the sort of

results I'd expect in PLoS Biology. That said, it may be that I'm missing the

big picture; if so, then I recommend the authors emphasize the importance

and/or significance of the results.

In more detail: all simulation results in the paper assume that sampling occurs

independently of location. This is sufficient to demonstrate the point that

WLDV is sensitive to number of samples, but does not demonstrate that the other

two metrics are robust to this aspect of sampling. In fact, I strongly suspect

that the other metrics (at least, WDC) are strongly affected by spatial

sampling. (Consider the extreme case where all samples came from nearly the

same location; then ancestral locations would also be inferred to be from that

same location as well, and so the inferred diffusivity would be very small.) To

be fair, the authors do not misrepresent their findings (e.g., the abstract

says ``robust to the number of samples''), and there is a paragraph in the

Discussion about this limitation. So, I think these results are useful and not

wrong, just lacking in depth.

The empirical results are also much harder to interpret given this caveat:

since all data is from other publications, there is no description of sampling,

so one cannot even guess whether spatial sampling might affect results.

(Ideally, there would be maps of sampling locations for each outbreak.) It may

well be that the results presented are not strongly affected by sampling, but

again, these are lacking in depth and thus don't make the impact they might

have otherwise.

Other points:

In the simulations, there is no phylogenetic inference step. This is because

``adding a genomic sequence simulation step would have also drastically

increased the computation time and resources required to conduct the

analyses''. (I assume this means that the Bayesian inference of phylogeny

would be resource-intensive, since simulating even large genomes on a tree is

trivial.) This may be a fair point, but if this paper is to be a comprehensive

assessment of the other two metrics, rather than just a takedown of WLDV, then

phylogenetic error should also be assessed (at least in a few simulations, or

perhaps in a non-Bayesian framework).

The mathematical explanation of why WLDV depends on sample size (around

l.147-158) is nice and simple but very awkwardly explained IMO: I think the

explanation could be simplified and smoothed out substantially. Also, it might

be good to relate this to the literature: a name for the observation on l.157

is that the "total variation of a Brownian path is infinite", and a name for

the observation on l.162 is that "Brownian motion is an infinitely divisible

process with constant quadratic variation". (However, these points aren't

going to help non-mathematical readers understand the argument.) Also note that

the equation on l.162 applies to any random walk with independent increments of

finite variance, not just Brownian, since the variance of the sum of

independent things is the sum of their variances.

The authors may want to cite Neigel and Avise, Genetics, 1993, who have the

same goal with essentially the same type of data.

Finally a note: the scheme of randomly rotating two independent Cauchys seems

fine to me; however for future reference a less ad hoc method of obtaining a

"bivariate Cauchy" is in my opinion to use the "scale mixture of Normals"

approach, i.e., to divide a standard bivariate Normal by the square root of a

Gamma with the appropriate parameters (search "Cauchy scale mixture of

Normals").

Minor comments:

l.21: "allowing to unveil": "unveil" is imprecise; also, the grammar is wrong ("allowing us to unveil" or "which can unveil" would be better)

l.22: "accuracy of dispersal insights" -> "accuracy of dispersal estimates" 

l.24: "implement a simulation framework" -> "use simulations"

l.52: "allows estimating" -> "allows estimation of"

l.69: "connected *by* the air traffic network", maybe?

l.73: "by essence" -> "in essence"

l.87: "on each phylogeny branch" -> "on the i-th branch of the phylogeny"

l.101-102: "longitudinal and latitudinal displacements are randomly drawn from a Cauchy distribution" - this is not accurate,

 because of the (important) random rotation later mentioned. Please rephrase.

l.103: "The example" -> "An example", maybe?

l.119: "We have then estimated the three dispersal metrics" - as defined, your metrics are quantities

 computed from data, and so are "computed", not "estimated". (They are statistics, not some underlying parameter

 you are trying to estimate.)

l.248: "relatively important diffusion coefficient" -> do you mean "large"?

l.275: "adding a genomic sequence simulation step would have also drastically increased the computation time and resources required to conduct the analyses"

 - perhaps clarify that the step that would take substantial time is that of Bayesian inference; generation of genome sequences is trivial

 (e.g., with msprime).

l.295: "the diffusion coefficient allows estimating a diffusion coefficient as the invaded area per unit of time" - awkward;

 also, although the expression involves distance-squared, I don't think the best interpretation of this is as "area'.

l.301: "between *infection locations of* two successive hosts"

l.308: "The comparative framework initiated in the present study" - the comparisons are excellent; the framework is not new.

l.320: "Unfortunately, fully Bayesian phylogeographic approaches" - why is it important for these analyses that the inference be fully Bayesian?

l.337: delete "Of note, we defined a sampling window so that"

l.338: "In between the two," - between the two what?

l.341: "in a normal distribution of" -> "from a Normal distribution with"

l.340 and l.342: the phrase "stochastically defined" is odd: better "sampled", maybe?

l.340: The description here is not correct, or at least confusing, since it

 first says that d_x and d_y are the horizontal and vertical displacements, but

 then it says that these are randomly rotated (and so hence aren't actually the

 horizontal and vertical displacements). It might be better to (a) omit the

 random rotation for the Normal (even if you did the random rotation in your

 code, since these are mathematically equivalent); and (b) for the Cauchy say

 you picked a random angle and then did two independent, orthogonal

 displacements in the directions defined by that angle.

l.343: "in a Cauchy distribution" -> "from a Cauchy distribution"

l.343: Unclear what the purpose of the citation is - explain.

l.347: "gradient" -> "direction"

Reviewer #2: 

The authors' efforts to directly address the impact of sampling heterogeneity is appreciated and is highly relevant to a major challenge to epidemiologic studies incorporating genomic sequencing data. However, the efforts to examine the robustness of various methods have been applied to animal data. This is understandable from a methodological standpoint - animal to animal transmission is more likely to be constrained by immediate geographic proximity, unlike humans who travel by air and other modes. However, the problem at hand (whether sampling depth impacts inference for these methods) seems to be a more relevant problem to human studies, particularly outbreak investigations where many connected cases are sampled and sequenced. Any further connection that can be made to applications in human data of this first component would strengthen the paper.

Secondly, the evaluations of geographic spread are interesting and valuable. These simulations, and the methods presented, are a helpful basis for future studies of phylogeography of viruses. These are particularly valuable for future studies of animal viruses that are emerging, for which methods to quickly detect changes in spread dynamics are needed.

---

## [Decision Letter · Decision Letter 2]

30 Sep 2024

Dear Dr Dellicour,

Thank you for your patience while we considered your revised manuscript "How fast are viruses spreading in the wild?" for consideration as a Research Article at PLOS Biology. Your revised study has now been evaluated by the PLOS Biology editors, the Academic Editor [and the original reviewers - EDIT AS APPLICABLE]. 

As you will see in the reports, reviewer #1 appreciates the improvement of the manuscript but still raises additional points that should be addressed. but Reviewer #1 still has some comments that should be addressed. Specifically, to address issues related to spatial sampling, the reviewer suggests providing maps of the sampling locations in the supplementary material and discussing more deeply how the sampling scheme may have influenced the results. Addressing these concerns is crucial for further consideration of your manuscript for publication in PLOS Biology.

**IMPORTANT - SUBMITTING YOUR REVISION**

*Resubmission Checklist*

*Published Peer Review*

*PLOS Data Policy*

*Blot and Gel Data Policy*

Sincerely,

Melissa

Melissa Vazquez Hernandez, Ph.D.

Associate Editor

PLOS Biology

REVIEWER'S COMMENTS:

Reviewer #1: 

Review of "How fast are viruses spreading in the wild?"

Thanks to the authors for the additional work put into this manuscript, which I

think has improved it. However, this additional work hasn't really changed my

initial opinion of the paper: I think it is a useful and interesting paper, but

some more depth and careful analysis would be necessary (in my view) to draw

strong conclusions from the results. I think this could be fixed with

relatively little additional background provided by the authors.

In more detail: the paper still does not meaningfully investigate the impact of

spatially heterogeneous sampling: there is a single plot that examines a single

example of spatially restricted sampling that's probably the least worrisome

possible situation. The paper has made the issue more obvious, including with

citations to other papers discussing the point, and I don't think anyone will

misconstrue this paper as concluding that the metrics are robust to changes in

spatial sampling. However, this is still a concern because the main conclusion

of the paper (see the title and abstract) has to do with the interpretation of

empirical datasets, for which there is absolutely no assessment or

communication of spatial sampling. I suspect that the authors have some good

reason to think that this is unlikely to substantially affect the results

(based on experience, perhaps?). A solution that would have made me satisfied

would be providing maps of sampling locations in the supplement and some

discussion of how sampling scheme might affect results. I am not suggesting

the authors perform an exhaustive evaluation of how various sampling schemes

affect these metrics, as that would be a whole new paper.

To be clear: spatial sampling *definitely does* affect all metrics computed

here, and many empirical datasets are strongly spatially biased -- for

instance, all samples from two large cities, or along one highway. Roughly

speaking, positive spatial autocorrelation of sampling probability should

reduce the estimated dispersal distance -- if, for instance, most samples came

from a few spatially restricted areas -- and negative spatial autocorrelation

should increase it -- if, for instance, available samples were thinned to avoid

sequencing lots of samples from the same location. The situation simulated in

the current paper almost entirely avoids either of these issues, as uniform

sampling in a large sub-range of the region shouldn't affect things much at

all. Again, perhaps this is not a problem for the empirical datasets used, but

in a paper whose topic is bias induced by sampling I would have expected

something more than a sentence in the Discussion.

At some point it would be good to point out - if I'm right on this point - that

WDC is not an unbiased estimator of the diffusion coefficient under the

Brownian model (since locations of internal nodes in the phylogeny are

estimated), and refer to something from phylogenetic comparative methods (a

paper of Felsenstein, probably?) for a method that would.

Abstract: "that can inform on" -> "that can be informative of" (or something)

Abstract: "recommendations for the *use of* lineage dispersal metrics"

Results: "can actually be expected" -> "is expected", perhaps?

Results: "quadratic distance" -> "squared distance"?

Results: "it seems that the random selection of sequences has often more impact

 on the dispersal metric estimates than the number of samples that were

 subsampled": My suspicion for why the different simulations in Figure S8

 (i.e., the distinct draws from the RRW process) look so different

 is because you're using the Cauchy distribution, and the same plot

 for the Brownian model would look much more consistent between replicates.

 (Also see Conclusion.)

Figure S8: Where are the results for the "scenario of sampling bias (SB)"?

---

## [Editor Report · Decision Letter 3]

23 Oct 2024

Dear Dr Dellicour,

Thank you for your patience while we considered your revised manuscript "How fast are viruses spreading in the wild?" for publication as a Research Article at PLOS Biology. This revised version of your manuscript has been evaluated by the PLOS Biology editors and the Academic Editor.

Based on our Academic Editor's assessment of your revision, we are likely to accept this manuscript for publication, provided you satisfactorily address the following data and other policy-related requests.

IMPORTANT - please attend to the following:

a) Please address my Data Policy requests below; specifically, we need you to supply the numerical values underlying Figs 1, 2, 3, S1, S2, S2, S3, S4, S5, S6, S7, S8, S9A-R, either as a supplementary data file or as a permanent DOI’d deposition. I note that you already have an associated GitHub deposition (https://github.com/sdellicour/dispersal_capacities); can you clarify that this contains the data underlying all the Figures? Also, because Github depositions can be readily changed or deleted, please make a permanent DOI’d copy (e.g. in Zenodo) and provide this URL (see below).

b) Please cite the location of the data clearly in all relevant main and supplementary Figure legends, e.g. “The data underlying this Figure can be found in S1 Data” or “The data underlying this Figure can be found in https://zenodo.org/records/XXXXXXXX

c) Please make any custom code available, either as a supplementary file or as part of your data deposition. I assume this will be in your Github/Zenodo deposition.

We expect to receive your revised manuscript within two weeks. 

*Published Peer Review History*

*Press*

Sincerely,

Roli Roberts

Roland G Roberts PhD

Senior Editor

PLOS Biology

rroberts@plos.org

on behalf of

Melissa Vazquez Hernandez, Ph.D.

Associate Editor

PLOS Biology

DATA POLICY:

[Figs….]

CODE POLICY

DATA NOT SHOWN?

---

## [Editor Report · Decision Letter 4]

27 Oct 2024

Dear Dr Dellicour,

Thank you for the submission of your revised Research Article "How fast are viruses spreading in the wild?" for publication in PLOS Biology. On behalf of my colleagues and the Academic Editor, Jonathan Dushoff, I am pleased to say that we can in principle accept your manuscript for publication, provided you address any remaining formatting and reporting issues. These will be detailed in an email you should receive within 2-3 business days from our colleagues in the journal operations team; no action is required from you until then. Please note that we will not be able to formally accept your manuscript and schedule it for publication until you have completed any requested changes.

IMPORTANT: I would strongly encourage you to make the peer-review process open. We believe that the readers could benefit of the reviewers' comments. 

PRESS

Sincerely,

Melissa 

Melissa Vazquez Hernandez, Ph.D., Ph.D.

Associate Editor

PLOS Biology
